# Comparing Single-Objective Optimization Protocols for Calibrating the Birds Nest Aquifer Model—A Problem Having Multiple Local Optima

**DOI:** 10.3390/ijerph17030853

**Published:** 2020-01-30

**Authors:** Richard T. Lyons, Richard C. Peralta, Partha Majumder

**Affiliations:** 1Department of Civil and Environmental Engineering, Utah State University, Logan, UT 84322-4110, USA; ricklyons@mac.com; 2College of Water Conservancy and Hydropower Engineering, Hohai University, Nanjing 211100, Jiangsu, China; parthamajp@gmail.com

**Keywords:** calibration, optimization, nonlinear optimization, groundwater, Grey Wolf Optimization, Particle Swarm Optimization, Meta-Heuristic Optimization, PEST calibration, Birds Nest Aquifer, Uinta Basin

## Abstract

To best represent reality, simulation models of environmental and health-related systems might be very nonlinear. Model calibration ideally identifies globally optimal sets of parameters to use for subsequent prediction. For a nonlinear system having multiple local optima, calibration can be tedious. For such a system, we contrast calibration results from PEST, a commonly used automated parameter estimation program versus several meta-heuristic global optimizers available as external packages for the Python computer language—the Gray Wolf Optimization (GWO) algorithm; the DYCORS optimizer framework with a Radial Basis Function surrogate simulator (DRB); and particle swarm optimization (PSO). We ran each optimizer 15 times, with nearly 10,000 MODFLOW simulations per run for the global optimizers, to calibrate a steady-state, groundwater flow simulation model of the complex Birds Nest aquifer, a three-layer system having 8 horizontal hydraulic conductivity zones and 25 head observation locations. In calibrating the eight hydraulic conductivity values, GWO averaged the best root mean squared error (RMSE) between observed and simulated heads—20 percent better (lower) than the next lowest optimizer, DRB. The best PEST run matched the best GWO RMSE, but both the average PEST RMSE and the range of PEST RMSE results were an order of magnitude larger than any of the global optimizers.

## 1. Introduction

Groundwater simulation models help engineers and scientists better understand water flow systems and potential effects of pumping and other diversions introduced into real-world systems. Model calibration is an essential step in demonstrating that a model can appropriately represent the real world. The process of calibrating a contaminant transport models first involves calibrating a flow model. PEST has been used extensively to reduce the effort and improve the quality of model calibrations [1,2]. PEST and PEST++ are available as a stand-alone command-line run computer program and some version of PEST is also integrated within several MODFLOW-based groundwater modeling software packages—Visual MODFLOW [3], Groundwater Vistas [4], Processing Modflow [5], and Groundwater Modeling System (GMS) [6].

Simulation model calibration can be viewed as an optimization problem. Global optimization techniques have long been considered and used for groundwater calibration [7,8]. The calibration goal is to determine the set of decision variable values that enable the model to produce output hydrologic estimates that best match values observed in the field—to produce the smallest root mean squared error (RMSE) between simulated and observed values. PEST uses the gradient search-based Gauss–Marquardt–Levenberg (GML) method to identify the set of decision variables that are the best parameter value inputs to the simulation model.

The GML method works well for identifying a locally optimal solution for a convex region of the solution space. However, the GML method cannot guarantee finding the globally optimal solution for a nonlinear solution space that has multiple local optima. [9,10,11].

To address the situation of multiple optima in preliminary modeling efforts, we made multiple runs, beginning each with a different initial guess of the optimal solution. Each initial guess converged to the nearest optimal solution. Comparison of the RMSEs of the different locally optimal solutions enabled selecting a solution that provided the lowest RMSE as a globally optimal solution. A challenge to this approach is that a user does not initially know how many optimal solutions exist. A PEST user can somewhat address this by making an exhaustive set of runs that use all practical combinations of maximum and minimum parameter values. As the number of parameters being calibrated increases, the modeler becomes less confident that they have identified all the most important optima. Because the global optimizers in this study randomly choose their starting input variables, instead of using an exhaustive set of runs, we created 15 sets of 8 randomly selected input parameters as our PEST initial values.

PEST is the de facto option for groundwater calibration, and is most commonly used. We hypothesized that optimization algorithms developed specially to address very nonlinear systems could be useful for calibration. Their use could potentially also decrease the complexity of calibration preparation. Because meta-heuristic optimization techniques avoid entrapment in local optima [12], here we evaluate use of several meta-heuristic algorithms for calibration.

Some recent groundwater studies suggest using newer algorithms for groundwater calibration and parameter estimation—Ensemble Kalman filtering [13]; null-space Monte Carlo [14]; Particle Swarm Optimization (PSO); Pattern Search [15]; Differential Evolution, Cat-Swarm Optimizer (CSO), and Particle Swarm Optimizer(PSO) [16]; CSO, PSO, and Grey Wolf Optimizer (GWO) [17]. 

The meta-heuristic PSO [18] introduced an innovative optimization method based on nature-based bird flocking and swarming theory. PSO has been used extensively in water resources engineering [19,20,21], and even as a calibration tool [15].

A newer swarm-based GWO furthers nature-based optimization by introducing a scheme based on the social hierarchy and hunting behavior of grey wolves. Compared with other swarm-based optimizers such as PSO, GWO has fewer adjustable parameters and allows wolves/search agents to approach, encircle and attack prey/optima [12]. GWO has been quickly adapted and used in many fields including water engineering, but has been used little in groundwater modeling [17,22]. 

Response surface methods of expressing an objective function value, such as radial basis function interpolation, are flexible, do not require derivatives, and can avoid entrapment in local optima, but are not as simple to implement as some meta-heuristic methods. Radial basis function interpolation is a well-known, mesh-free, response surface model [23] and has been used in groundwater engineering [24]. The DYCORS optimization framework was developed to use radial basis functions in highly-dimensional, black-box processes. The DYCORS/radial basis function (DRB) uses a dynamic search strategy that perturbs only a subset of the current best solution until an optimal solution is found [25].

The discussed PSO, GWO, and DRB optimizers are available as third-party black-box optimizer packages with the Python computer language. A high-level, easy to learn language, Python is easily expanded by third-party packages that are not included in the base Python language. This feature allows ready access to pre-built optimizers that are beyond the scope of the general Python writers and maintainers. It allows engineers and scientists to access and use codes that might otherwise be difficult and time-consuming to reproduce.

In this calibration comparison we use PSO, GWO, and DRB optimizers through Python, and compare results with the results from PEST. Although these meta-heuristic optimization algorithms are not new, their easy implementation through Python can allow them to effectively complement PEST use. Comparing results might encourage use of the optimizers for calibrating models of very nonlinear systems, especially groundwater systems. 

## 2. Materials and Methods

The goal of this study is to compare the use of three metaheuristic optimizers versus PEST for calibrating horizontal hydraulic conductivity values for a Birds Nest Aquifer (BNA) simulation model. In collaboration with the State of Utah, Division of Oil, Gas, and Mining, this study is a step toward future modeling that will evaluate horizontal flow patterns resulting from potential water injection. Calibration runs to date employed a personal computer with Windows 10 operating system. We ran PEST using the command line interface. Preliminary PEST tests involving changing the PEST default options did not noticeably affect processing time or the computed RMSE. Therefore, all PEST runs subsequently reported here used PEST default options.

We ran PSO, GWO, and DRB in Python using third-party packages—Pyswarm [26], SwarmPackagePy [27], and pySOT [23]. We used PSO by utilizing the default options of Pyswarm. We implemented GWO through SwarmPackagePy. We employed DRB through PySOT (Python Surrogate Optimization Toolbox). Of PySOT’s options, we used the radial basis function surrogate model (default), Symmetric Latin hypercube experimental design, and DYCORS search strategy [25].

Table 1 illustrates that as general input, the above global optimizers require the number of dimensions (number of decision variables to be optimized), upper and lower bounds on decision variables, maximum number of allowable iterations, and number of optimizer agents. Each optimizer uses a different equation for the total number of MODFLOW simulations to run, based on agent size and number of iterations. The total number of GWO simulations equals the number of agents times the number of iterations plus two times the number of agents plus two times the number of iterations. So, 30 agents and 300 iterations yields 30 × 300 + 2(30) + 2(300) = 9660. Total number of PSO simulations equals the number of agents times the number of iterations plus the number of agents. So, 30 agents and 321 iterations yields 30 × 321 + 30 = 9660. Total number of DRB simulations is merely the number of iterations. An agent is one set of candidate solutions, consisting of all decision variables. So, for a model with 8 dimensions, an optimizer using 30 agents would use 30 sets, each having 8 decision variables. GWO automatically performs the full number of iterations and therefore does not require any further inputs. We used default values for the additional inputs that PSO requires: particle velocity scaling factor (0.5), scaling factors to control search away from the best positions of each particle and the swarm (0.5, 0.5), minimum step size before search termination (1 × 10^−8^), and minimum change of swarm’s best objective value before search termination(1 × 10^−8^). We used default values for the additional inputs required by the DRB surrogate model: a kernel object (CubicKernel), a polynomial tail object (LinearTail(8)), and a regularization parameter (1 × 10^−6^). 

We coded a Python script to run our calibration. The script loads the appropriate optimizer inputs into each optimizer. The optimizer creates a group of decision variables that the Python script writes to a MODFLOW input file. The script then instructs MODFLOW to run, producing an output file of well heads. The Python-scripted objective function file reads the well heads and calculates a fitness value that is read by the optimizer. This process continues until the maximum number of iterations are run or a search termination criterion is satisfied. The optimizer then reports the decision variables and the fitness value of the best result. The process is shown in Figure 1. 

### Site

The roughly 700 square mile BNA (Figure 2), is located in the Uinta Basin in north-eastern Utah. This Eocene era aquifer lies beneath the unconfined Uintah and Duchesne aquifers. BNA base lies at depths beneath the ground surface ranging from about 210 feet to nearly 5800 feet. The BNA is recharged primarily by assumedly uniform seepage from the Uinta aquifer, and by seepage mainly from Evacuation Creek in the south east [28]. Discharge, mainly via a spring near Bitter Creek, follows a path probably facilitated by a Gilsonite vein [28]. The three-layers of the aquifer consist mainly of organic-lean oil shale. The top and bottom layers are differentiated from the middle layer by the existence of abundant saline crystals and large to medium nahcolite (NaHCO3) crystals that, in many cases, have been voided through dissolution increasing the secondary porosity. Outcrop studies have shown numerous faults and fissures. Thus, this generally tight aquifer has increased secondary permeability and porosity. Core studies have suggested that hydraulic conductivity might be higher at locations 14X-34, 14-36, X-13, 17, 4, and Corehole 2, due to significant to extensive dissolution of the nahcolite and saline minerals. Location 42-34 might have high conductivity due to several minor thin silty beds that could increase permeability. Corehole 9, Skyline 16, EX-1, and Red Wash 1 show only partial dissolution that might or might not significantly increase conductivity. Utah State 1 has no signs of dissolution [29]. Drawdown tests on three observation wells produced a 0.08 feet/day to 118.1 feet/day range of average horizontal conductivity [30].

BNA is represented in MODFLOW as a three-layer 3-D model with 42 rows and 97 columns. Each cell is a half mile long per side. Seventeen constant head cells simulate discharge and recharge. There are 25 head observation locations. All layers are confined and vertical conductivity is proportional to horizontal conductivity. The current BNA model has eight horizontal hydraulic conductivity zones, designated by HK_#. 

This paper focuses on calibrating the horizontal hydraulic conductivities for the eight zones existing in the current BNA model. Based upon data from core studies and drawdown tests, our first draft BNA model had about 50 horizontal conductivity zones. Further guided by that data and preliminary calibration runs, we combined most zones that had similar conductivities. The range of horizontal conductivities reported from pump tests serve as the upper and lower bounds on values considered during the calibration runs reported here. 

Our calibration comparison employed 15 runs per combination of optimizer and number of agents (preliminary 15 run-trials per optimizer captured a wide range of results for each optimizer). Each combination run performed approximately 9660 MODFLOW simulations. The number of agents (size of the agent pool) for each run was either 30 or 100. There is no consensus on how many agents to use for an optimizer. For this effort, 30 and 100 agents give an adequate range of results. The five-character name assigned to a combination consists of the three-digit optimizer abbreviation, followed by the number of agents.

## 3. Results

Table 2 shows that for 15 runs of the tested methods, GWO produced the hydraulic conductivity value sets that yielded the best simulated head—best (lowest) root mean squared error (RMSE), lowest average RMSE, and narrowest range of RMSEs. A narrow RMSE range indicates that there was little difference in RMSE produced by each run. As shown later, the GWO runs also produced small ranges in values of each of the hydraulic conductivity values. In this case, it might be possible to use any single GWO optimizer run, knowing that its results will not be significantly different than any other GWO run, including outliers. Whereas, if using a PSO optimizer, it would be wise to make multiple optimizer runs because the conductivity values estimated by different runs can be statistically significantly different. PEST matched GWO for the best run, but the PEST average RMSE and range were significantly larger than all global optimizers, by an order of magnitude. 

The average processing times of the different global optimizer runs in Python were very similar. However, PEST, a local optimizer, required many fewer MODFLOW simulations than the global optimizers. The Figure 3 plots of RMSE improvement with iteration for the best run of each method, shows that initially, PEST obtained better results in fewer simulations, followed by DRB with 100 agents and then PSO and GWO with 30 agents. However, as the number of simulations increased, GWO matched the best final RMSE of PEST.

The average optimizer performance, shown in Figure 4, highlights that in DRB and GWO, the number of agents does not change the results much. Generally, DRB appears to quickly resolve to its optimum and then, on average, produces no better fitness over time. However, the running improvements of best run shows that DRB fitness might slowly improve. GWO resolves nearly as quickly as DRB, but continues to improve continuously, resulting in a better final fitness. Initially, PSO with 30 agents performs similarly to GWO, but does not resolve to as good a fitness. Overall, performance of PSO lags behind that of DRB and GWO.

PEST matched the best RMSE in one run, and did so much more rapidly, than any other method, but the average PEST run performance is worse than any of the global optimizers. Nevertheless, PEST ran an average of merely 278 MODFLOW simulations in 15 runs, and more runs might have yielded better results (The results of an extra 15 random runs in PEST confirmed that the first 15 runs had explored the search space effectively, without getting a better solution). All the global optimizers ran about 10,000 simulations, but only GWO continuously improved throughout the entire set of simulations. PSO’s RMSE was about double of that of GWO and only continued to improve from about half to three-fourths of the total number of simulations. DRB appears to be the most efficient of the global optimizers, because it achieved a low RMSE in relatively few simulations, with a low average RMSE that was better than PSO. However, GWO consistently produced the lowest RMSE.

The Figure 5 boxplots display the range and average of RMSE results from tested optimizer combinations and PEST. The GWO30 combination provided the lowest mean, 4.25 ft/d, and the smallest range, 0.12 ft/d, of fitness results and had a low standard deviation of 0.04. The results from DRB100 were quite good as well with a mean of 5.61 ft/d, a range of 1.15 ft/d, and standard deviation of 0.29. Results from the best PSO option, PSO30, had a mean double of that of either GWO result, and a range about 70 times larger than GWO30 and over 7 times larger than DRB100. PEST exhibits, by far, the largest range and the highest mean. PEST can perform well, but is not as consistent as the global optimizers.

Table 3 shows the horizontal conductivity (HK) values from the best runs. For all 15 runs, Table 4 shows the HK standard deviation of each zone. Notably, the values for both GWO and DRB are very small in zones HK_2, HK_3, and HK_4. Zone conductivity standard deviations from PSO and PEST do not vary much in magnitude. All conductivity values presented by PEST and the optimizers were within the imposed upper and lower bounds that we derived from field studies [30]. In fact the low conductivity zone agrees with a previous work modeling study [28]. Our reported HK values, potentiometric surface map, and resulting flow directions are reasonable with respect to previous BNA reports and core studies [28,30]. The HK_1 conductivity reported by PEST, and the HK_5 conductivity reported by PSO30 were at the posed upper bounds. GWO, PSO, and DRB do not compute or report a sensitivity analysis. PEST reports sensitivities that identify parameter changes that most significantly change the resulting RMSE. PEST reports normalized sensitivities for each iteration and composite sensitivities for the entire optimization process. PEST normalizes by dividing each sensitivity value by the number of model observations that it is trying to match. 

## 4. Conclusions

We compared the results from three Python-based optimizer packages—Grey-Wolf (GWO), particle swarm (PSO), and DYCORS/radial basis function (DRB)—and PEST, when calibrating for eight horizontal conductivity values of the three-layer Birds Nest Aquifer. In terms of computed root mean squared error (RMSE) of simulated head, both GWO and DRB performed as well as, or better, than PEST. Preparation of input data and running GWO is simple. Although DRB has more options requiring input, all preset default options worked well. 

Overall, GWO provided the best RMSE, regardless of whether using 30 or 100 agents. Although 100 agents performed better in early simulations, as simulations continued, the results from GWO using 30 and 100 agents became similar. The RMSE values computed by 15 GWO/30 agent runs have the narrowest range of any tested method. The second and third narrowest ranges came from GWO/100 agents and DRB/100 agents, respectively. The narrow range in RMSE results adds confidence that any single GWO optimization would produce good results. Even more important, the set of conductivity values determined by GWO/30 agents are very similar to those by the best PEST results. The range of RMSE results from PEST runs is much larger than any of the global optimizers. Although for this site, PEST did match the best fitness provided by any of the global optimizers, for another more nonlinear calibration problem, a PEST user might be uncertain that they had obtained close to a globally optimal solution. 

Our work highlights the promise of the recently developed GWO nature-based optimizer in terms of simplicity of implementation and goodness of results. Further investigations using GWO might promote its use in more fields and applications. It might also be valuable to perform similar comparisons for more complex scenarios that have a much greater dimensionality or more complex interaction between parameters. That additional comparison could test the suitability and the limits of each optimizer.

## Figures and Tables

**Figure 1 ijerph-17-00853-f001:**
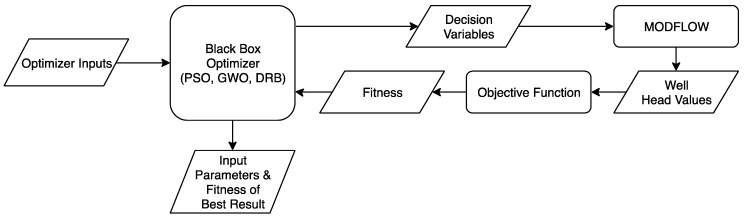
Flowchart of calibration process for the global optimizers. PSO: Particle Swarm Optimization; GWO: Gray Wolf Optimization; DRB: DYCORS/radial basis function.

**Figure 2 ijerph-17-00853-f002:**
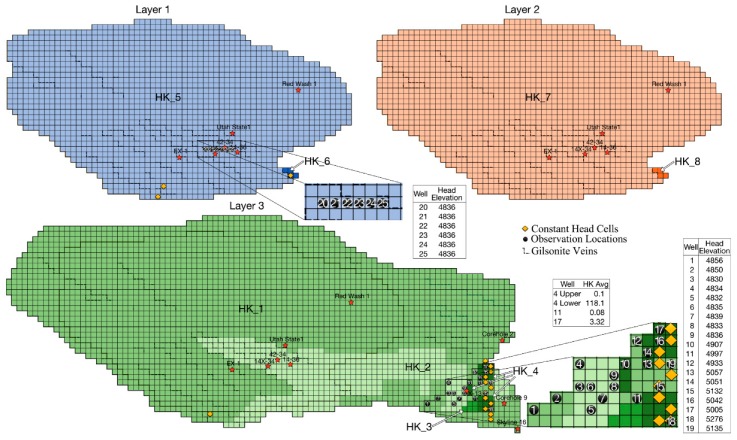
Three-layer Birds Nest Aquifer (BNA) model constant head cells, observation locations, layers, and horizontal conductivity zones (indicated by different shades and HK_#). Inset closeups of observation well locations and adjacent tables of observed head elevations. Inset table of average conductivity estimates from well drawdown tests.

**Figure 3 ijerph-17-00853-f003:**
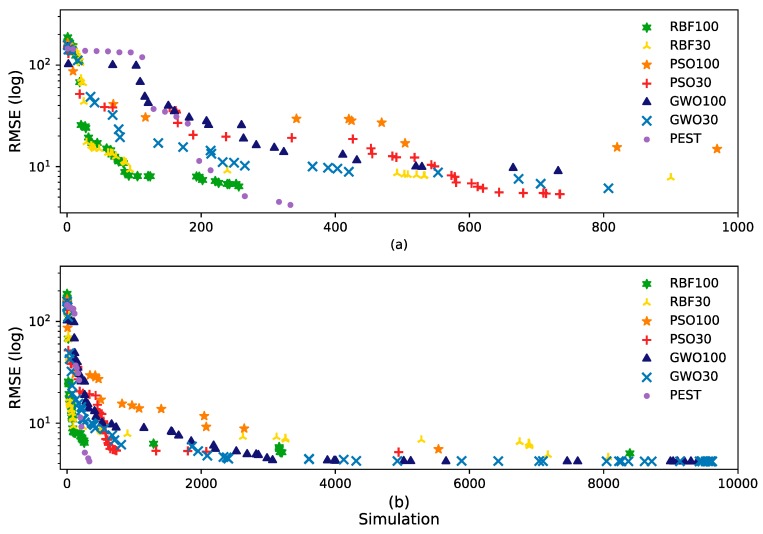
RMSE improvements with simulation of the best run of each method—(**a**) 1000 simulations and (**b**) 10,000 simulations.

**Figure 4 ijerph-17-00853-f004:**
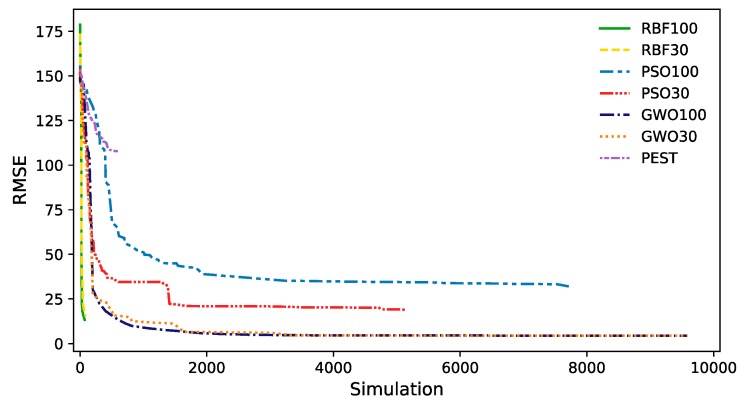
Running improvements of the mean of 15 runs by optimizer/agent size pairing.

**Figure 5 ijerph-17-00853-f005:**
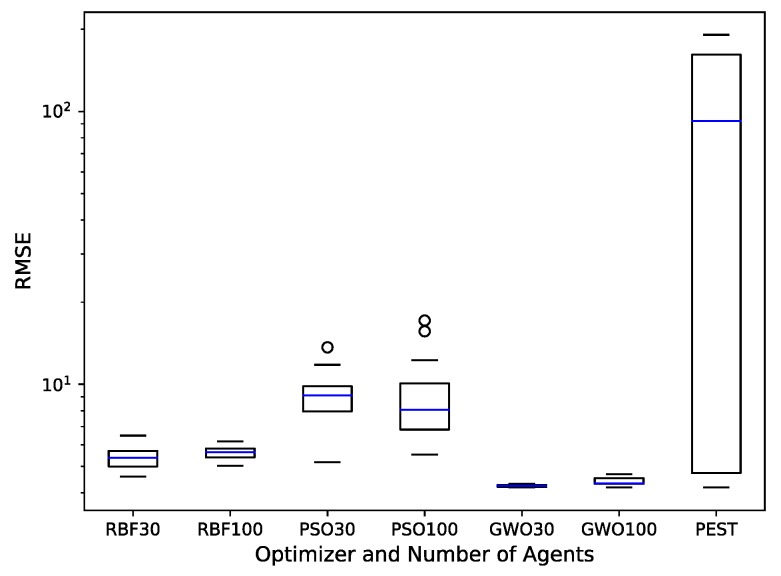
Box plot of the root mean squared error (RMSE) produced by 15 runs of each optimizer and PEST.

**Table 1 ijerph-17-00853-t001:** The three global optimizers, their respective Python packages, common inputs needed and additional inputs. PSO: Particle Swarm Optimization; GWO: Gray Wolf Optimization; DRB: DYCORS/radial basis function.

Optimizer:	PSO	GWO	DRB
Python Package:	Pyswarm	SwarmPackagePy	PySOT
Common Inputs:	Number of dimensions; Upper and lower bounds; Max number of iterations; Number of optimizer agents		
Additional Inputs:	Scaling factors: particle velocity, search away from best position of each particle, search away from best position of swarm; minimum step size before search termination; minimum change of swarm’s best objective value before search termination		Kernel object; polynomial tail object; regularization parameter;
Initial Strategy:	PSO chooses decision variables randomly within the set bounds.	GWO chooses decision variables randomly within the set bounds.	DRB chooses decision variables based on a Symmetric Latin Hypercube design.
Number of MODFLOW simulations:	agents(1+iterations)	iterations * agents + 2(iterations + agents)	iterations
Search Methods:	Nature-based bird flocking and swarming theory	Nature-based social hierarchy and hunting behavior of grey wolves	Surrogate Model, perturbs only a subset of current best solution

**Table 2 ijerph-17-00853-t002:** Average and minimum (best) results from 15 runs from each pairing of optimizer and number of agents (best result in bold green, second best in orange).

Variables	DRB100	DRB30	PSO100	PSO30	GWO100	GWO30	PEST
Time (s) Avg	2955	3005	3082	3197	3010	3053	**114**
Time (s) Min	2918	2955	2066	2507	1768	1762	**12**
RMSE (ft/d) Avg	5.61	5.39	9.14	8.93	4.41	**4.25**	88.82
RMSE (ft/d) Min	5.03	4.59	5.52	5.18	**4.19**	4.19	4.19
σ	0.29	0.53	3.36	2.04	0.14	0.04	20.43
Range (ft/d)	1.15	1.90	11.60	8.48	0.50	0.12	186.55
Iteration no. of last RMSE Improvement	8390	8068	5537	4943	9569	9618	333

**Table 3 ijerph-17-00853-t003:** RMSE and horizontal conductivity (HK_#) values for the best run of each optimizer/agent number pair. Similar values per HK zone are colored the same.

	**RMSE**	**HK_1**	**HK_2**	**HK_3**	**HK_4**	**HK_5**	**HK_6**	**HK_7**	**HK_8**
DRB100	5.03	37.97	0.05	0.41	0.11	78.69	60.64	61.01	49.33
DRB30	4.59	69.08	0.06	0.44	0.14	74.70	12.15	17.77	0.34
PSO100	5.52	97.31	0.02	50.63	0.62	88.09	116.36	8.52	42.07
PSO30	5.18	33.90	0.05	0.32	0.11	118.00	66.45	45.38	92.64
GWO100	4.19	86.96	0.06	0.45	0.14	45.94	6.88	0.02	0.19
GWO30	4.19	117.73	0.06	0.45	0.15	1.78	2.13	11.08	0.20
PEST	4.19	118.00	0.06	0.45	0.14	0.32	0.83	11.27	0.22
Upper Bound		118.00	118.00	118.00	118.00	118.00	118.00	118.00	118.00
Lower Bound		0.01	0.01	0.01	0.01	0.01	0.01	0.01	0.01

**Table 4 ijerph-17-00853-t004:** Standard deviation of 15 runs of each HK zone by optimizer/agent number pair.

	**HK_1**	**HK_2**	**HK_3**	**HK_4**	**HK_5**	**HK_6**	**HK_7**	**HK_8**
DRB100	18.27	0.017	0.30	0.067	32.90	35.80	19.03	18.88
DRB30	18.73	0.023	0.24	0.11	28.86	37.87	34.73	30.93
PSO100	21.89	21.05	29.30	43.75	37.04	35.73	26.13	37.06
PSO30	31.38	12.86	39.05	37.54	39.36	39.57	33.49	32.66
GWO100	27.21	0.0010	0.007	0.0030	39.39	5.53	32.66	2.50
GWO30	0.21	0.0010	0.011	0.0022	19.14	3.44	32.18	3.70
PEST	47.69	24.78	41.70	25.00	39.17	21.45	46.65	34.45

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
