# Peer review of "Comparing Single-Objective Optimization Protocols for Calibrating the Birds Nest Aquifer Model—A Problem Having Multiple Local Optima"

_ijerph, 2020, doi:10.3390/ijerph17030853_

Round 1

Reviewer 1 Report

Need more water levels (observation) information to show the variation in aquifers and space. Need a table show the initial condition(cases) and searching methods (ex. surrogate model, search strategy) among different optimal methods with the description through line 89~105. That will make your comparisons clearer. The titles, ex. RBF”100” and RBF”30”, should explain again. Need other figures to present the less interactions (<4000) and less time of different methods. Line 173 and Table 2, I cannot see the information and relationship between NK_# and aquifer layers, please add more information. Add more reference, field data is better, to support the results of Table 2.

Author Response

Reviewer 1 comments.

“Need more water levels (observation) information to show the variation in aquifers and space. Need a table show the initial condition(cases) and searching methods (ex. surrogate model, search strategy) among different optimal methods with the description through line 89~105. That will make your comparisons clearer. The titles, ex. RBF”100” and RBF”30”, should explain again. Need other figures to present the less interactions (<4000) and less time of different methods. Line 173 and Table 2, I cannot see the information and relationship between NK_# and aquifer layers, please add more information. Add more reference, field data is better, to support the results of Table 2.” 

1a. Need more water levels (observation) information to show the variation in aquifers and space.

We expanded the map of BNA to include observation water elevations.

1b. Need a table show the initial condition(cases) and searching methods (ex. surrogate model, search strategy) among different optimal methods with the description through line 89~105. That will make your comparisons clearer.

We added the following Table description and diagram:

Table 1. The three global optimizers, their respective Python packages, common inputs needed and additional inputs.

Optimizer:

PSO

GWO

DRB

Python Package:

Pyswarm

SwarmPackagePy

PySOT

Common Inputs:

Number of dimensions; Upper & lower bounds; Max number of iterations; Number of optimizer agents

Additional Inputs:

Scaling factors: particle velocity, search away from best position of each particle, search away from best position of swarm; minimum step size before search termination; minimum change of swarm’s best objective value before search termination

Kernel object; polynomial tail object; regularization parameter

Initial Strategy:

PSO chooses decision variables randomly within the set bounds

GWO chooses decision variables randomly within the set bounds

DRB chooses decision variables based on a Symmetric Latin Hypercube design.

Number of MODFLOW simulations:

agents(1+iterations)

iterations * agents +

2(iterations + agents)

iterations

Search Methods:

Nature-based

bird flocking & swarming theory

Nature-based social hierarchy & hunting behavior of grey wolves

Surrogate Model, perturbs only a subset of current best solution

We coded a Python script to run our calibration. The script loads the appropriate optimizer inputs into each optimizer. The optimizer creates a group of decision variables that that the Python script writes to a MODFLOW input file. The script then instructs MODFLOW to run, producing an output file of well heads. The Python-scripted objective function file reads the well heads and  calculates a fitness value that is read by the optimizer. This process continues until the maximum number of iterations are run or a search termination criterion is satisfied. The optimizer then reports the decision variables and the fitness value of the best result. The process is shown in Figure 1.

Figure 1. Flowchart of calibration process for the global optimizers.

1c. The titles, ex. RBF”100” and RBF”30”, should explain again.

In line 180, we added, “The five-character name assigned to a combination consists of the three-digit optimizer abbreviation, followed by the number of agents.”

Table 2. text states “optimizer and number of agents” and Table 3. States “optimizer/agent number”

1d. Need other figures to present the less interactions (<4000) and less time of different methods.

Changed Figure 3 to add a pane showing only improvements up to 1000 iterations.

Figure 3. RMSE improvements with iteration of the best run of each method—(a) 1000 simulations and (b) 10,000 simulations.

1e. Line 173 and Table 2, I cannot see the information and relationship between NK_# and aquifer layers, please add more information.

We have expanded the figure of the Birds Nest Aquifer. It now includes HK zones labeled,  inset detail of the observation wells along with tables which show the measured head elevation, and the results of drawdown tests with estimated average hydraulic conductivity. We include an explanation of this material in lines 142-158. And added two new paragraphs lines 159-169.

1f. Add more reference, field data is better, to support the results of Table 2. 

In the updated Birds Nest Aquifer figure we added the calculated estimates of horizontal hydraulic conductivity at the only sites with any published information. We added a reference for the upper and lower bounds in the text along with reference lines 157-158. We also highlight the locations where geological cores are available and have explained in the text that it supports the HK values we see in the results, except in three locations on the edges of the study area.We added lines 237-241, “Zone conductivity standard deviations from PSO and PEST do not vary much in magnitude. All conductivity values presented by PEST and the optimizers were within the imposed upper and lower bounds that we derived from field studies [30]. In fact the low conductivity zone agrees with previous work modeling study[28]. Our reported HK values, potentiometric surface map, and resulting flow directions are reasonable with respect to previous BNA reports and core studies [28,30].”

Reviewer 2 Report

This paper utilized various algorithms to calibrate the Birds Nest Aquifer (BNA) model. Although the results seem to be encouraging, this paper is still lacking in many areas.

The introduction discusses current methods and algorithms used in the paper. However, the relations of these current methods with groundwater simulation or Birds Nest Aquifer are minimal. PSO, GWO, Response surface method, and RBF are not entirely new methods. Please elaborate on the state-of-the-art research in this area. In addition, please add another paragraph discussing the merit of this paper, the importance of the research presented, or border impacts on groundwater modeling. Section 2 is titled Materials and Methods. This has to be reorganized. The materials used are not clear. The methods are had to follow. Providing figures or flow charts of the methods would be helpful for readers to understand. What are the steps in creating the site model? What are the steps involved in the calibrations? Is there anything new in the methods? Also, some of these questions can be answered in the Method section. Why 30 agents and 100 agents? Why 15 runs? Are 15 runs sufficient? What are the convergence criteria?  In section 3, the results are obvious. However, it can be improved further by discussing the relationships of the results to the site model or the groundwater simulation challenges. For examples: - Line 163-166. What does the fitness do to the BNA model? -Why HK analysis is necessary? What is the relationship between HK with the site? How are the zones defined? What are the impacts of a “more sensitive” zone on the BNA model? There is a lot of information that is sacrificed in this paper.

Other minor comments:

The title does not highlight the presented work. Please consider improving the title.  Computer time is more commonly referred to as “computation or computational time”.

Author Response

Reviewer 2:

The introduction discusses current methods and algorithms used in the paper. However, the relations of these current methods with groundwater simulation or Birds Nest Aquifer are minimal. PSO, GWO, Response surface method, and RBF are not entirely new methods. Please elaborate on the state-of-the-art research in this area. In addition, please add another paragraph discussing the merit of this paper, the importance of the research presented, or border impacts on groundwater modeling. Section 2 is titled Materials and Methods. This has to be reorganized. The materials used are not clear. The methods are had to follow. Providing figures or flow charts of the methods would be helpful for readers to understand. What are the steps in creating the site model? What are the steps involved in the calibrations? Is there anything new in the methods? Also, some of these questions can be answered in the Method section. Why 30 agents and 100 agents? Why 15 runs? Are 15 runs sufficient? What are the convergence criteria?  In section 3, the results are obvious. However, it can be improved further by discussing the relationships of the results to the site model or the groundwater simulation challenges. For examples: - Line 163-166. What does the fitness do to the BNA model? -Why HK analysis is necessary? What is the relationship between HK with the site? How are the zones defined? What are the impacts of a “more sensitive” zone on the BNA model? There is a lot of information that is sacrificed in this paper.
Other minor comments:

The title does not highlight the presented work. Please consider improving the title.  Computer time is more commonly referred to as “computation or computational time”.

2a. the relations of these current methods with groundwater simulation or Birds Nest Aquifer are minimal. 

We added  sentences in lines 98-99 at the beginning of the Materials and Methods section that states this study is used for calibrating the BNA. We elaborate on this idea in a paragraph in the site section lines 162-167.

2b. PSO, GWO, Response surface method, and RBF are not entirely new methods.

In lines 40-41 we added acknowledgement and citation that using global optimization techniques for calibration is not new. We also added a final paragraph to the introduction, lines 92-96, that acknowledges that these optimizers are not new, but reiterates that “their easy implementation through Python can allow them to effectively complement PEST use. Comparing results might encourage use of the global optimizers for calibrating models of very nonlinear systems, especially groundwater systems.”

2c. Please elaborate on the state-of-the-art research in this area.

We have added in line 36 PEST++ to PEST in the first paragraph of the introduction. We have also added in line 61 that “PEST is the de facto option for groundwater calibration, and is most commonly used.” In the paragraph that follows, lines 66-69 cites more recent studies in groundwater calibration and parameter estimation, “Some more recent groundwater studies suggest using newer algorithms for groundwater calibration and parameter estimation—Ensemble Kalman filtering [13]; null-space Monte Carlo[14];Particle Swarm Optimization (PSO), Pattern Search [15]; Differential Evolution, Cat-Swarm Optimizer (CSO), Particle Swarm Optimizer(PSO) [16]; CSO, PSO, Grey Wolf Optimizer (GWO)[17].”

2d. In addition, please add another paragraph discussing the merit of this paper, the importance of the research presented, or border impacts on groundwater modeling.

We assume “border impacts” means “broader impacts”. Our results cause us to be more confident that we have the best calibration if we are using GWO instead of PEST. We added a short paragraph at the end of the introduction, lines 92-96, “In this calibration comparison we use PSO, GWO, and DRB optimizers through Python, and compare results with the results from PEST. Although these meta-heuristic optimization algorithms are not new, their easy implementation through Python can allow them to effectively complement PEST use. Comparison results might encourage use of the optimizers for calibrating models of very nonlinear systems, especially groundwater systems.”

.

2e. Section 2 is titled Materials and Methods. This has to be reorganized. The materials used are not clear. The methods are had to follow. Providing figures or flow charts of the methods would be helpful for readers to understand.

To clarify, we made the first line of the Materials and Methods section to be, “The goal of this study is to compare the ability of various optimizers and PEST to calibrate the horizontal hydraulic conductivity of the Birds Nest Aquifer.”We also rewrote much of this section and

changed wording in the introduction and methods. Specificaly we defined the term ‘decision variables’ and use that in lieu of some of the ‘input variables or input parameter’. In the Methods section, we added explanation for a new table 1. We added the following table paragraph and chart to help clarify

Table 1. The three global optimizers, their respective Python packages, common inputs needed and additional inputs.

Optimizer:

PSO

GWO

DRB

Python Package:

Pyswarm

SwarmPackagePy

PySOT

Common Inputs:

Number of dimensions; Upper & lower bounds; Max number of iterations; Number of optimizer agents

Additional Inputs:

Scaling factors: particle velocity, search away from best position of each particle, search away from best position of swarm; minimum step size before search termination; minimum change of swarm’s best objective value before search termination

Kernel object; polynomial tail object; regularization parameter

We coded a Python script to load the necessary inputs into each optimizer. We also created a Python objective function file for the black-box optimizers to use. The objective function file includes commands to read the optimizer-produced decision variables and to feed those into a MODFLOW input file. MODFLOW is then run. The resulting well head files from the MODFLOW output are used to calculate a fitness value that is sent to the optimizer. This process continues until the maximum number of iterations are run or a search termination criterion is satisfied. The optimizer then reports the decision variables and the fitness value of the best result. The process is shown in Figure 1.

Figure 1. Flowchart of calibration process for the global optimizers.

2f. What are the steps in creating the site model?

We added hydrogeologic field information in the site model section and references to core studies and the few drawdown tests for data. We replaced the the former 3D BNA figure with three 2D maps to include observation locations and observed conductivity and head values.

2g. What are the steps involved in the calibrations?

The new figure above in question 2e explains the steps more clearly.

2h. Is there anything new in the methods?

We added a final paragraph in the Introduction lines 92-96, r stating  that these optimizers widely useful for management optimization, might also be very valuable if used for the reverse problem of calibrating in groundwater hydrology and engineering, instead of the most common practice of using PEST.

2i. Why 30 agents and 100 agents?

We added text at the end of the site section, lines 175-177. “The number of agents (size of the agent pool) for each run was either 30 or 100. There is no consensus on how many agents to use for an optimizer. For this effort, 30 and 100 agents gives an adequate range of results.”

2j. Why 15 runs? Are 15 runs sufficient?

Added text in site section, lines 173-174, “Our calibration comparison employed 15 runs per combination of optimizer and number of agents (preliminary 15 run-trials per optimizer captured a wide range of results for each optimizer).” We also added, “(The results of an extra 15 random runs in PEST confirmed that the first 15 runs had explored the search space effectively, without getting a better solution).”

2k. What are the convergence criteria?

We added the values in lines 121-127

2l. In section 3, the results are obvious. However, it can be improved further by discussing the relationships of the results to the site model or the groundwater simulation challenges. For examples: - Line 163-166. What does the fitness do to the BNA model

Lines 238-241 explain “ All conductivity values presented by PEST and the optimizers were within the imposed upper and lower bounds that we derived from field studies. Our reported HK values, potentiometric surface map, and resulting flow directions are reasonable with respect to previous BNA reports and core studies [28,30].”

2m. -Why HK analysis is necessary?

Added to the beginning of the Materials and Methods, line 100, that “In collaboration with the State of Utah, Division of Oil, Gas and Mining, this study is a step toward future modeling that will evaluate horizontal flow patterns resulting from potential water injection.”

2n. What is the relationship between HK with the site?

In the Site description section, we included discussion how HK changes with nahcolite crystal dissolution. We added figures showing the HK zones in each layer. We added lines 235-240 “All conductivity values presented by PEST and the optimizers were within the imposed upper and lower bounds that we derived from field studies [30]. In fact the low conductivity zone agrees with previous work modeling study[28]. Our reported HK values, potentiometric surface map, and resulting flow directions are reasonable with respect to previous BNA reports and core studies [28,30].”

2o. How are the zones defined?

Added this to the text in site section, lines 163-165. The zones were defined during the modeling process. Based on field data that we now include and cite in the paper, our first version of the model had about about 50 zones and “Further guided by that data and preliminary calibration runs, we combined most zones that had similar conductivities.”

2p. The title does not highlight the presented work. Please consider improving the title. 

We added the site’s name and changed the title to be more representative.

Round 2

Reviewer 2 Report

All comments were addressed properly by the authors. No further comments.